# From Nonlinear Dominant System to Linear Dominant System: Virtual Equivalent System Approach for Multiple Variable Self-Tuning Control System Analysis

**DOI:** 10.3390/e25010173

**Published:** 2023-01-15

**Authors:** Jinghui Pan, Kaixiang Peng, Weicun Zhang

**Affiliations:** School of Automation and Electrical Engineering, University of Science and Technology Beijing, Beijing 100083, China

**Keywords:** virtual equivalent system, stochastic multivariable STC, stability, convergence

## Abstract

The stability and convergence analysis of a multivariable stochastic self-tuning system (STC) is very difficult because of its highly nonlinear structure. In this paper, based on the virtual equivalent system method, the structural nonlinear or nonlinear dominated multivariable self-tuning system is transformed into a structural linear or linear dominated system, thus simplifying the stability and convergence analysis of multivariable STC systems. For the control process of a multivariable stochastic STC system, parameter estimation is required, and there may be three cases of parameter estimation convergence, convergence to the actual value and divergence. For these three cases, this paper provides four theorems and two corollaries. Given the theorems and corollaries, it can be directly concluded that the convergence of parameter estimation is a sufficient condition for the stability and convergence of stochastic STC systems but not a necessary condition, and the four theorems and two corollaries proposed in this paper are independent of specific controller design strategies and specific parameter estimation algorithms. The virtual equivalent system theory proposed in this paper does not need specific control strategies, parameters and estimation algorithms but only needs the nature of the system itself, which can judge the stability and convergence of the self-tuning system and relax the dependence of the system stability convergence criterion on the system structure information. The virtual equivalent system method proposed in this paper is proved to be effective when the parameter estimation may have convergence, convergence to the actual value and divergence.

## 1. Introduction

The stability and convergence analysis of stochastic self-tuning control systems is more difficult than that of deterministic self-tuning control systems. It is difficult to analyze and understand in theory, which makes it very difficult for engineers and technicians to analyze the stability and convergence of such systems in practice.

References [1,2,3,4] studied the stability and convergence of the self-tuning control system consisting of the minimum variance control strategy and the stochastic gradient parameter estimation algorithm. References [5,6] provided the stability and convergence results of the self-tuning control system consisting of the minimum variance control strategy and the least squares parameter estimation algorithm. References [7,8] presented the stability and convergence results of the self-tuning control system consisting of the pole-placement control strategy and the weighted least squares parameter estimation algorithm. As a summary of the stability and convergence of STC, the results regarding the minimum variance control strategy do not require parameter estimation convergence; however, the results regarding other control strategies, such as pole placement, require parameter estimation convergence [9,10].

The above results are all for the minimum phase object, and the common feature is that the convergence of parameter estimation is not required. If the controlled object is of a non-minimum phase, the self-tuning algorithm of the minimum variance type cannot be used for control, but pole assignment and other control strategies need to be used. The corresponding stability and convergence analysis are more difficult than the adaptive control system of the minimum variance type. The existing results are basically obtained under the premise of parameter estimation convergence (can converge to the real value or non-real value); there are also some results that do not require the convergence of parameter estimation but can only guarantee the stability and robustness of the system, and cannot guarantee the convergence of the system.

For the controlled object with non-minimum phase certainty, [11,12] proved the stability and convergence of pole placement self-tuning control by introducing additional excitation signals. The stability and convergence of pole assignment self-tuning control can also be obtained by modifying the estimation model without introducing an additional excitation signal [13]. The literature [1,14] analyzes the stability and convergence of the pole assignment algorithm with a “key technology lemma”. For non-minimum phase random controlled objects, refs. [15,16] adopted the method of adding an “attenuated excitation signal” to ensure the stability and convergence of random pole assignment self-tuning control, while [7,8] provided that it does not need an external excitation signal, but uses a self-convergent weighted least squares parameter estimation algorithm to ensure the stability and convergence of random pole assignment self-tuning control. The literature [17,18,19,20] analyzed the stability and convergence of the adaptive decoupling control system, and the literature [21] analyzed the stability and convergence of generalized minimum variance self-tuning control for minimum phase objects and some non-minimum phase objects based on the Lyapunov function. The literature [22] proposed a theory of virtual equivalent systems, but it mainly focused on single-variable systems and did not study multivariable systems.

The disadvantage of the minimum variance self-tuning control method is that it is not suitable for non-minimum phase objects. The main reason is that the unstable pole of the regulator cannot be exactly canceled with the zero point of the object, resulting in the instability of the system. In addition, even if the generalized minimum variance self-tuning controller is used, in order to ensure the closed-loop stability of the system, the control weight factor is usually determined through trial and error. This constraint also introduces significant inconvenience to specific applications. The computation amount of the random gradient algorithm is much less than that of the least squares algorithm, but its convergence speed is very slow. Moreover, under the conditions of strong, persistent excitation, the parameter estimation error of the system using the stochastic gradient algorithm converges to zero uniformly, but under other conditions, it is very difficult to prove the convergence of the stochastic gradient algorithm. The least squares estimation method is simple in its algorithm and is easy to implement. It does not need to know the statistical information of the measurement error, but its accuracy is difficult to improve. Its limitations are reflected in two aspects: first, it can only estimate the deterministic constant value vector but cannot estimate the time process of the random vector; second, it can only ensure the minimum mean square error of the measurement, but it does not ensure the best estimation error of the estimator, and the accuracy is not high. The stochastic self-tuning system using the pole placement method requires high accuracy of the model, has the problem of modeling error, and requires the convergence of parameter estimation.

In view of the characteristics and shortcomings of the above control algorithms, this paper, based on the theory of virtual equivalent systems, weakens the conditions required by the stability and convergence criteria of the stochastic self-tuning system, mainly eliminating the direct dependence on the order information of the controlled object, reducing the requirements for parameter estimation errors, and eliminating the dependence of the pole placement self-tuning control strategy on the convergence of parameter estimation, The difficulty of analysis is transferred from the system structure to the compensation signal, thus reducing the difficulty of the original problem.

New self-tuning control schemes are still emerging, such as the robust multi-model adaptive control system, fuzzy parameter self-tuning PID method, intelligent AC contactor self-tuning control technology, self-tuning control method of simulation turntable based on the accurate identification of the model parameters, sliding model adaptive control method [23,24,25,26,27], and the traditional approach cannot prove its stability and convergence due to the lack of a general theory of adaptive control. In general, it is expected that the stability and convergence analysis methods of stochastic STC systems are independent of specific control strategies and parameter estimation methods. Some scholars have made efforts in this field to develop a general theory [28,29,30,31], but the results are not satisfactory.

There are many related achievements that are difficult to enumerate one by one. In recent years, many adaptive control schemes related to stability and convergence have achieved good results in practical applications [32], but there are no theoretical analysis results.

New adaptive control schemes are emerging, and it is difficult to analyze the stability and convergence of each adaptive control system one by one. For this reason, people have been expecting to find a unified stability and convergence analysis method [23,33,34,35,36]. However, despite some sporadic results [32,34,37,38], the expected unified analysis method and theory still need to be explored. The concept of the virtual equivalent system and its corresponding analysis methods are generated under such a background [25,39,40,41]. Weicun Zhang, one of the authors of this paper, proposed the concept of the virtual equivalent system and then analyzed the stability and convergence of various self-tuning control systems in a unified framework, converting the nonlinear system into an equivalent linear system with an infinitesimal nonlinear compensation signal. 

In this paper, we will consider three cases of parameter estimation: (1) The parameter estimation converges to the real value; (2) the parameter estimation converges to the non-real value; (3) and the parameter estimation may not converge. The second and third cases do not require the structural information of the plant. Considering the particularity of the minimum variance self-tuning control, a criterion with an intuitive explanation is given for this kind of STC system.

It is worth pointing out that for a general multiple-input multiple-output stochastic self-tuning control system, not only the minimum variance control system but also the convergence of parameter estimation is not a necessary condition for the stability and convergence of the STC system.

Through the theoretical and experimental research in this paper, it is concluded that for the self-tuning control system of nonlinear controlled objects (deterministic or stochastic, minimum phase or non-minimum phase), to ensure its stability and convergence, only the boundedness of parameter estimation, slow time variation and the output approximation effect of the estimation model (i.e., the parameter estimation error is relatively infinitesimal) are required, and the control strategy meets the stability and tracking requirements according to the principle of deterministic equivalence.

## 2. Virtual Equivalent System of Stochastic Self-Tuning Control System

For the convenience of description, we first consider the following multivariable stochastic system ΣP with known structural information but unknown parameters (for the discussion of general stochastic systems containing colored noise, see Section 4).
(1)$$\Sigma P:A(q^{-1})y(k)=q^{-d}B(q^{-1})u(k)+\omega (k)$$
where, y(k), u(k) and ω(k), are the output signals, the input signal and the noise signal with the appropriate dimension of the plant to be controlled, respectively. 

Assuming that
y(k)=0,u(k)=0,ω(k)=0,∀k<0,
(2)$$\underset{n\to \infty }{\lim}\frac{1}{n}\sum_{i=1}^{n}{\left\|\omega (i)\right\|}^{2}=R<\infty ,\;a.s.$$
(3)$$\begin{array}{l} A(q^{-1})=I+A_{1}q^{-1}+\ldots +A_{n}q^{-n},n\geq 1\\ B(q^{-1})=B_{0}+B_{1}q^{-1}+\ldots +B_{m}q^{-m},m\geq 1 \end{array}$$

Introducing symbols
$$\theta^{T}=[-A_{1},\ldots -A_{n},B_{0},..B_{m}],$$
$$\phi^{T}(k-d)=[y(k-1),\ldots y(k-n),\ldots ,u(k-d),..u(k-d-m)],$$

Then, we have
(4)$$y(k)=\phi^{T}(k-d)\theta +\omega (k)$$

The estimated model is denoted as $\Sigma P_{m}(k)$, and its parameter matrix is
$$\Sigma P_{m}(k):{\hat{\theta }}^{T}(k)=\left[-{\hat{A}}_{1},\ldots ,-{\hat{A}}_{n},{\hat{B}}_{0},..,{\hat{B}}_{m}\right]$$

The performance of the parameter estimation can be expressed by the (posterior) model output error
(5)$$e(k)=y(k)-\phi^{T}(k-d)\hat{\theta }(k)-\omega (k)=\phi^{T}(k-d)\theta_{0}-\phi^{T}(k-d)\hat{\theta }(k).$$

The self-tuning controller is denoted by ΣC(k), and can also be treated as a matrix $\theta_{c}(k)$. The controller can be obtained by various design methods, such as pole placement. Different control strategies actually represent different mapping, i.e.,
$$f:\hat{\theta }(k)\to \theta_{c}(k),\;\mathrm{or}\;\theta_{c}(k)=f(\hat{\theta }(k)),$$

Additionally, the control law is generally denoted by
$$u(k)=\phi_{c}^{T}(k)\theta_{c}(k),$$
where
$$\phi_{c}{}^{T}(k)=[y_{r}(k),y_{r}(k-1)\ldots y(k),y(k-1)\ldots ,u(k-1),\ldots ].$$
where $y_{r}(k)$ is a known bounded reference signal. 

The above-described self-tuning control system is shown in Figure 1, which is abbreviated as $\left(\Sigma C(k),\Sigma P\right)$.

Accordingly, the real plant corresponds to an ‘ideal’ controller, i.e.,
$$\left\{ \begin{array}{l} \theta_{c}=f(\theta )\\ u_{0}(k)=\phi_{c}^{T}(k)\theta_{c} \end{array}.\right.$$

This constant control system is abbreviated as (ΣC,ΣP),as shown in Figure 2. On the basis of (ΣC,ΣP), we can artificially construct a system that is equivalent in the input–output sense to the self-tuning control system. It consists of the constant control system of Figure 2 and a compensational signal Δu(k), abbreviated as $\left(\Sigma C,\Sigma P,\Delta u(k)\right)$, as shown in Figure 3.
(6)$$\Delta u(k)=u(k)-u_{0}(k)=\phi_{c}^{T}(k)\theta_{c}(k)-\phi_{c}^{T}(k)\theta_{c}.$$

Since $\theta_{c}$ is unknown, Δu(k) cannot be calculated exactly, but it can be estimated (see the analysis below for details). 

The above equivalent system is recorded as the “virtual equivalent system” of the self-tuning control system. The reason why it is recorded as “virtual” is that it exists but is unknown. One of the merits of the “virtual equivalent system” is that it quantitatively reflects the difference between a self-tuning control system and the corresponding constant control system. The definition of the convergence of the self-tuning control system is based on the constant system shown in Figure 2.

The stability of the self-tuning control system is defined by
$$\underset{n\to \infty }{\lim}\sup\frac{1}{n}\sum_{i=1}^{n}\left({\left\|y(i)\right\|}^{2}+{\left\|u(i)\right\|}^{2}\right)<\infty .$$

The convergence of the self-tuning control system is defined by
$$\underset{n\to \infty }{\lim}\frac{1}{n}\sum_{i=1}^{n}{\left\|\left(y(i)-y_{r}(i)\right)\right\|}^{2}=\underset{n\to \infty }{\lim}\frac{1}{n}\sum_{i=1}^{n}{\left\|\left(y_{1}(i)-y_{r}(i)\right)\right\|}^{2}.$$

## 3. Main Results

### 3.1. Parameter Estimation Converges to the True Value

Considering a self-tuning control system based on an arbitrary control strategy and an arbitrary parameter estimation algorithm, the following results are obtained.

**Theorem** **1.**
*For the self-tuning control system of (1), if the following conditions are met.*

*The parameter estimation converges to a true value.*
*The control strategy satisfies the stability requirements of the object with known parameters, then the close- loop system composed by* $\left(\Sigma P,\Sigma C\right)$*is stable.**The mapping* $f\left(\cdot \right)$*is continuous at* $\hat{\theta }(k)=\theta$.
*Then, the self-tuning control system is stable and convergent.*


**Proof.** The stability of the system is proved by the method of contradiction, and then the convergence is proved. Since the virtual equivalent system shown in Figure 3 is a linear constant structure (its time-varying nonlinear features are transferred to Δu(k)), it can be decomposed into two subsystems, one is the constant control system shown in Figure 2, and the other is the system shown in Figure 4.
(7)$$y(k)=y_{1}(k)+y_{2}(k),$$
(8)$$u(k)=u_{1}(k)+u_{2}(k).$$By the superposition principle and considering the situation of the two subsystems separately, the system shown in Figure 2 is a conventional random control system. The second condition in Theorem 1 ensures that it is closed-loop stable, so there is
(9)$$\underset{n\to \infty }{\lim}\sup\frac{1}{n}\sum_{i=1}^{n}\left({\left\|y_{1}(i)\right\|}^{2}+{\left\|u_{1}(i)\right\|}^{2}\right)<\infty ,$$
(10)$$\underset{n\to \infty }{\lim}\frac{1}{n}\sum_{i=1}^{n}{\left\|\left(y_{1}(i)-y_{r}(i)\right)\right\|}^{2}<\infty .$$For the system shown in Figure 4, there is no influence of noise due to the closed-loop system being stable, so we have [42]
(11)$$\sum_{k=1}^{n}||y_{2}(k)|{|}^{2}\leq M_{1}\sum_{k=1}^{n}{\left\|\Delta u(k)\right\|}^{2}+M_{2}\;,\;0<M_{1}<\infty ,\;0\leq M_{2}<\infty ,$$
(12)$$\sum_{k=1}^{n}||u_{2}(k)|{|}^{2}\leq M_{3}\sum_{k=1}^{n}{\left\|\Delta u(k)\right\|}^{2}+M_{4}\;,\;0<M_{3}<\infty ,\;0\leq M_{4}<\infty ,$$That is,
$$\sum_{k=1}^{n}||y_{2}(k)|{|}^{2}=O\left(\sum_{k=1}^{n}{\left\|\Delta u(k)\right\|}^{2}\right)+M_{2}\text{,}\sum_{k=1}^{n}||u_{2}(k)|{|}^{2}=O\left(\sum_{k=1}^{n}{\left\|\Delta u(k)\right\|}^{2}\right)+M_{4}.$$By theorem condition (1) and condition (3) in Theorem 1, we have $\theta_{c}(k)\to \theta_{c}$, $\left\|\Delta u(k)\right\|=o(\left\|\phi_{c}(k)\right\|)$.By the composition of $\phi_{c}(k)$, we know that $\left\|\phi_{c}(k)\right\|=O(\left\|\phi (k-d)\right\|)+M$, M is a bounded constant.Furthermore, by the convergence of parameter estimation, we have
$$\frac{1}{n}\sum_{k=1}^{n}{\left\|\Delta u(k)\right\|}^{2}=o(\frac{1}{n}\sum_{k=1}^{n}\left(\frac{1}{n}\sum_{k=1}^{n}{\left\|\phi (k-d)\right\|}^{2}\right)).$$Thus,
(13)$$\frac{1}{n}\sum_{k=1}^{n}||y_{2}(k)|{|}^{2}=o\left(\frac{1}{n}\sum_{k=1}^{n}{\left\|\phi (k-d)\right\|}^{2}\right)$$
(14)$$\frac{1}{n}\sum_{k=1}^{n}||u_{2}(k)|{|}^{2}=o\left(\frac{1}{n}\sum_{k=1}^{n}{\left\|\phi (k-d)\right\|}^{2}\right)$$Then, to prove the following formula
$$\underset{n\to \infty }{\lim}\sup\frac{1}{n}\sum_{i=1}^{n}\left({\left\|y(i)\right\|}^{2}+{\left\|u(i)\right\|}^{2}\right)<\infty .$$It suffices to prove that
$$\underset{n\to \infty }{\lim}\sup\frac{1}{n}\sum_{i=1}^{n}{\left\|\phi (k-d)\right\|}^{2}<\infty .$$We can construct $\phi_{1}(k-d)$ (corresponding to the system of Figure 2) and $\phi_{2}(k-d)$ (corresponding to the system of Figure 4), so that
$$\phi (k-d)=\phi_{1}(k-d)+\phi_{2}(k-d).$$It can be seen from Equations (13) and (14) that
$$\frac{1}{n}\sum_{i=1}^{n}{\left\|\phi_{2}(k-d)\right\|}^{2}=o(\frac{1}{n}\sum_{k=1}^{n}{\left\|\phi (k-d)\right\|}^{2})$$By the triangle inequalities, we have
(15)$$\frac{1}{n}\sum_{i=1}^{n}{\left\|\phi (k-d)\right\|}^{2}=\frac{1}{n}\sum_{i=1}^{n}{\left\|\phi_{1}(k-d)+\phi_{2}(k-d)\right\|}^{2}\;\leq \frac{1}{n}\sum_{i=1}^{n}{\left\|\phi_{1}(k-d)\right\|}^{2}+\frac{1}{n}\sum_{i=1}^{n}{\left\|\phi_{2}(k-d)\right\|}^{2}\;=\frac{1}{n}\sum_{i=1}^{n}{\left\|\phi_{1}(k-d)\right\|}^{2}+o(\frac{1}{n}\sum_{k=1}^{n}{\left\|\phi (k-d)\right\|}^{2}).$$Furthermore, considering
$$\underset{n\to \infty }{\lim}\sup\frac{1}{n}\sum_{i=1}^{n}\left({\left\|y_{1}(i)\right\|}^{2}+{\left\|u_{1}(i)\right\|}^{2}\right)<\infty$$Thus, we obtain
$$\underset{n\to \infty }{\lim}\sup\frac{1}{n}\sum_{i=1}^{n}{\left\|\phi_{1}\left(k-d\right)\right\|}^{2}<\infty .$$Taking (15) into consideration, we obtain
$$\underset{n\to \infty }{\lim}\sup\frac{1}{n}\sum_{i=1}^{n}{\left\|\phi (k-d)\right\|}^{2}<\infty .$$Thus, combining the above formula with (13) and (14), it follows that
(16)$$\frac{1}{n}\sum_{k=1}^{n}||y_{2}(k)|{|}^{2}=o(1),$$
(17)$$\frac{1}{n}\sum_{k=1}^{n}||u_{2}(k)|{|}^{2}=o(1).$$Next, we prove
$$\underset{n\to \infty }{\lim}\frac{1}{n}\sum_{i=1}^{n}||y(i)-y_{r}(i)|{|}^{2}=\underset{n\to \infty }{\lim}\frac{1}{n}\sum_{i=1}^{n}||y_{1}(i)-y_{r}(i)|{|}^{2}.$$Combining Cauchy’s Inequality with (10) and (16), we have
$$0\leq {\left\{\frac{1}{n}\sum_{i=1}^{n}\left\|y_{1}(i)-y_{r}(i)\right\|\cdot \left\|y_{2}(i)\right\|\right\}}^{2}\leq \{\frac{1}{n}\sum_{i=1}^{n}{\left\|y_{1}(i)-y_{r}(i)\right\|}^{2}\}.\{\frac{1}{n}\sum_{i=1}^{n}{\left\|y_{2}(i)\right\|}^{2}\}\to 0.$$By the Squeeze Theorem, we obtain
$$\underset{n\to \infty }{\lim}{\left\{\frac{1}{n}\sum_{i=1}^{n}\left\|y_{1}(i)-y_{r}(i)\right\|\cdot \left\|y_{2}(i)\right\|\right\}}^{2}=0.$$It follows that
(18)$$\underset{n\to \infty }{\lim}\frac{1}{n}\sum_{i=1}^{n}\left\|y_{1}(i)-y_{r}(i)\right\|\cdot \left\|y_{2}(i)\right\|=0.$$Finally, let us consider
$$\underset{n\to \infty }{\lim}\frac{1}{n}\sum_{i=1}^{n}{\left\|y(i)-y_{r}(i)\right\|}^{2}=\underset{n\to \infty }{\lim}\frac{1}{n}\sum_{i=1}^{n}{\left\|\left(y_{1}(i)-y_{r}(i))+y_{2}(i\right)\right\|}^{2}.$$According to the norm triangle inequality and (16) and (18), we have
$$\begin{array}{ll} \frac{1}{n}\sum_{i=1}^{n}{\left\|\left(y_{1}(i)-y_{r}(i))+y_{2}(i\right)\right\|}^{2} & \leq \frac{1}{n}\sum_{i=1}^{n}{\left\|y_{1}(i)-y_{r}(i)\right\|}^{2}+\frac{1}{n}\sum_{i=1}^{n}{\left\|y_{2}(i)\right\|}^{2}+\frac{2}{n}\sum_{i=1}^{n}\left\|y_{1}(i)-y_{r}(i)\right\|\cdot \left\|y_{2}(i)\right\|\\ & \to \frac{1}{n}\sum_{i=1}^{n}{\left\|y_{1}(i)-y_{r}(i)\right\|}^{2} \end{array}.$$Similarly,
$$\begin{array}{ll} \frac{1}{n}\sum_{i=1}^{n}{\left\|\left(y_{1}(i)-y_{r}(i))+y_{2}(i\right)\right\|}^{2} & \geq \frac{1}{n}\sum_{i=1}^{n}{\left\|y_{1}(i)-y_{r}(i)\right\|}^{2}+\frac{1}{n}\sum_{i=1}^{n}{\left\|y_{2}(i)\right\|}^{2}-\frac{2}{n}\sum_{i=1}^{n}\left\|y_{1}(i)-y_{r}(i)\right\|\cdot \left\|y_{2}(i)\right\|\\ & \to \frac{1}{n}\sum_{i=1}^{n}{\left\|y_{1}(i)-y_{r}(i)\right\|}^{2} \end{array}.$$According to the Squeeze Theorem, we have
$$\underset{n\to \infty }{\lim}\frac{1}{n}\sum_{i=1}^{n}{\left\|\left(y_{1}(i)-y_{r}(i))+y_{2}(i\right)\right\|}^{2}=\underset{n\to \infty }{\lim}\frac{1}{n}\sum_{i=1}^{n}{\left\|y_{1}(i)-y_{r}(i)\right\|}^{2},$$
i.e.,
$$\underset{n\to \infty }{\lim}\frac{1}{n}\sum_{i=1}^{n}{\left\|y(i)-y_{r}(i)\right\|}^{2}=\underset{n\to \infty }{\lim}\frac{1}{n}\sum_{i=1}^{n}{\left\|y_{1}(i)-y_{r}(i)\right\|}^{2}.$$That completes the proof of Theorem 1. □

### 3.2. Parameter Estimation Converges to Non-True Value

Considering the fact that the structure information of the controlled object is unknown, the order of the estimated model can be lower than the order of the real controlled object, which is often the case in practical engineering applications.

**Theorem** **2.**
*For the self-tuning control system of the plant (1), if*
*(1)* *The parameter estimate converges to*$\;\theta_{0}$
, *the estimated model*
$\;\Sigma P_{m}(k)$
*is consistently controllable*,
$$\sum_{k=1}^{n}{\left\|y(k)-\phi^{T}(k-d)\hat{\theta }(k)-\omega (k)\right\|}^{2}=o(1+\sum_{k=1}^{n}{\left\|\phi (k-d)\right\|}^{2}).$$*(2)* 
*The control strategy satisfies the stability requirements for the known objects of the parameters;*
*(3)* *The mapping*f(⋅)*is continuous at*$\hat{\theta }(k)=\theta_{0}$.
*Then, the self-tuning control system is stable and convergent.*



**Proof.** The stability of the system is proved by the method of contradiction, and then the convergence is proved. In order to prove Theorem 2, we need to build another virtual equivalent system, as shown in Figure 5.$\Sigma P_{0}$ represents the model corresponding to the convergence value $\theta_{0}$ of the parameter estimation, and $\Sigma C_{0}$ represents the controller corresponding to $\Sigma P_{0}$.The virtual equivalent system shown in Figure 5 is different from the virtual equivalent system shown in Figure 3. First, $\Sigma C_{0}$ and $\Sigma P_{0}$ are different from ΣC and ΣP, respectively. Second, the definitions of $e^{\prime }(k)$ and $\Delta u^{\prime }(k)$ are different from (5) and (6), respectively. In Figure 5$e^{\prime }(k)=y(k)-\phi^{T}(k-d)\theta_{0}=y(k)-\phi^{T}(k-d)\hat{\theta }(k)+\phi^{T}(k-d)\hat{\theta }(k)-\phi^{T}(k-d)\theta_{0}$, is
(19)$$e^{\prime }(k)=e(k)+\phi^{T}(k-d)\hat{\theta }(k)-\phi^{T}(k-d)\theta_{0},$$(20)$$\Delta u^{\prime }(k)=u(k)-u_{0}(k)=\phi_{c}^{T}(k)\theta_{c}(k)-\phi_{c}^{T}(k)\theta_{c0}.$$It is known from condition (1) in Theorem 2.
(21)$$\sum_{k=1}^{n}{\left\|e(k)-\omega (k)\right\|}^{2}=o(1+\sum_{k=1}^{n}{\left\|\phi (k-d)\right\|}^{2}).$$Then, we have
$$e^{\prime }(k)-\omega (k)=e(k)-\omega (k)+\phi^{T}(k-d)\hat{\theta }(k)-\phi^{T}(k-d)\theta_{0}.$$It is also known by condition (1) that $\hat{\theta }(k)\to \theta_{0}$, so we have
(22)$$\sum_{k=1}^{n}{\left\|e^{\prime }(k)-\omega (k)\right\|}^{2}=o(1+\sum_{k=1}^{n}{\left\|\phi (k-d)\right\|}^{2}).$$Further, from condition (3), we have
(23)$$\left\|\Delta u^{\prime }(k)\right\|=o(\beta +\left\|\phi (k-d)\right\|).$$Thus, Δu(k) has the same properties as in the proof of Theorem 1, i.e.,
$$\frac{1}{n}\sum_{k=1}^{n}{\left\|\Delta u(k)\right\|}^{2}=o(\frac{1}{n}\sum_{k=1}^{n}{\left\|\phi (k-d)\right\|}^{2}).$$Decomposing the system shown in Figure 5 into three subsystems (as shown in Figure 6, Figure 7 and Figure 8, respectively), it is known from condition (2) that the subsystem, as shown in Figure 6, is stable; and the rest of the proof process is similar to that of Theorem 1 (details are omitted to save space). □

### 3.3. Parameter Estimation May Not Converge

This section demonstrates two theorems for the STC, consisting of the minimum variance control strategy and the arbitrary control strategy. As described in Section 3.2, the structural information of the estimated model could be inconsistent with the real control plant.

First, let us consider the minimum variance control strategy and explain why this particular type of self-tuning control system does not require the convergence of parameter estimation. 

**Theorem** **3.**
*For the minimum variance type self-tuning control system of plant (1), any feasible parameter estimation algorithm can be used if the following conditions are met.*
*(1)* $B\left(q^{-1}\right)$*is Hurwitz stable polynomial, and*$\left|B_{0}\right|\neq 0$.*(2)* 
*Control strategy*

 u(k)

*exists.*
*(3)* 
*Parameter estimation satisfies.*


$\sum_{k=1}^{n}{\left\|e(k)-\omega (k)\right\|}^{2}=o(\alpha +\sum_{k=1}^{n}{\left\|\phi (k-d)\right\|}^{2})$

*,*

α

*is a non-zero constant.*

*The self-tuning control system is then stable and convergent.*



**Proof.** Using the virtual equivalent system shown in Figure 3, under the condition that condition (2) and condition (3) are satisfied, it can be proved that the minimum variance self-tuning control has the following special properties [22].
(24)$$\Delta u(k)=B_{0}{}^{-1}[\phi^{T}(k-1)(\theta_{0}-\hat{\theta }(k))].$$Further,
$$\Delta u(k)=B_{0}{}^{-1}[e(k)-\omega (k)].$$Therefore, Δu(k) has the following property
$$\frac{1}{n}\sum_{k=1}^{n}{\left\|\Delta u(k)\right\|}^{2}=o(\frac{1}{n}\sum_{k=1}^{n}{\left\|\phi (k-d)\right\|}^{2}).$$Decompose the virtual equivalent system of the minimum variance self-tuning control system into two subsystems, as shown in Figure 3 and Figure 4. The rest of the proof process is similar to that of Theorem 1, and the details are omitted.In fact, the key to the proof process of the three theorems is to prove the property of Δu(k) or $\Delta u^{\prime }(k)$ in the virtual equivalent system. In Theorems 1 and 2, considering the arbitrary (linear) control strategy, the mapping relationship between the estimated parameters and the controller parameters is complicated. Therefore, parameter estimation convergence is required to ensure the properties of Δu(k). In the minimum variance control strategy, the controller parameters can be directly represented by the estimated parameters. Additionally, we have $\Delta u(k)=B_{0}{}^{-1}[e(k)-\omega (k)]$; therefore, parameter estimation convergence is not required. Only$\sum_{k=1}^{n}{\left\|e(k)-\omega (k)\right\|}^{2}=o(\alpha +\sum_{k=1}^{n}{\left\|\phi (k-d)\right\|}^{2})$ is needed to ensure the stability and convergence of the minimum variance controller. □

**Corollary** **1.**
*Considering a minimum-variance type self-correcting control system using any feasible parameter estimation algorithm, if*
*(1)* $B(q^{-1})$*is a Hurwitz stable polynomial, and*$\left|B_{0}\right|\neq 0$.*(2)* 
*Control law*

u(k)

*exists.*
*(3)* 
*The parameter estimation error is bounded; that is,*

$\frac{1}{n}\sum_{k=1}^{n}{\left\|e(k)-\omega (k)\right\|}^{2}\leq M^{\prime }<\infty$

*, then the self-tuning control system is stable.*



The stability and convergence of a self-tuning control system consisting of an arbitrary control strategy when parameter estimation may not converge are considered below.

**Theorem** **4.**
*Self-tuning control system for the controlled plant (1), if*
*(1)* 

$\left\|\hat{\theta }(k)\right\|\leq M<\infty$

*,*

$\left\|\hat{\theta }(k)-\hat{\theta }(k-l)\right\|\to 0$

*,*

l

*is a finite value.*
*(2)* 

$\sum_{k=1}^{n}{\left\|e(k)-\omega (k)\right\|}^{2}=o(\alpha +\sum_{k=1}^{n}{\left\|\phi (k-d)\right\|}^{2})$

*,*

α

*is a non-zero constant.*
*(3)* 
*The control strategy satisfies the stability requirements for the known parameters and tracks the reference signal*
*(4)* *The controller parameter is a continuous function of the parameter estimates; that is,*$\theta_{c}(k)$*is a continuous function of*$\hat{\theta }(k)$. 
*If the above conditions are met, the self-tuning control system is stable and convergent.*



**Remark** **1.**
*To ensure that the parameter estimates are bounded, a projection approach can be used, see references [43,44,45,46].*


**Proof.** Considering another virtual equivalent system, as shown in Figure 9, where $P_{m}(k)$ and C(k) are corresponding to $\Sigma P_{m}(k)$ and ΣC(k), respectively.The system shown in Figure 9 is further converted into the virtual equivalent system shown in Figure 10. From conditions (1) and (4) of Theorem 4, the interval between $t_{k}$ and $t_{k-1}$ can be chosen to be sufficiently large, such that to maintain the property of the required $\Delta u^{\prime }(k)$. Therefore, the system as shown in Figure 10 is a “slow switching” system.Next, the virtual equivalent system shown in Figure 10 is decomposed into three subsystems, as shown in Figure 11, Figure 12 and Figure 13, respectively. Figure 11 is a stochastic system. Figure 12 and Figure 13 are deterministic systems. By conditions (1) and (2), we have
$$\sum_{k=1}^{n}{\left\|e_{i}(k)-\omega (k)\right\|}^{2}=o(\alpha +\sum_{k=1}^{n}{\left\|\phi (k-d)\right\|}^{2}).$$Based on the results of the “slow switching” stochastic system [43,44,45,46] and conditions (1) and (3), it is known that the system shown in Figure 11 is stable and tracking. The rest of the proof process is similar to that of Theorem 1.Further, considering the low-order modeling situation, we have the following result.**Corollary** **2.**
*Consider a self-tuning control system consisting of any feasible parameter estimation algorithm and control strategy if the following conditions hold.*
*(1)* $\left\|\hat{\theta }(k)\right\|\leq M<\infty$, $\left\|\hat{\theta }(k)-\hat{\theta }(k-l)\right\|\to 0$, l*is a finite value.**(2)* $\frac{1}{n}\sum_{k=1}^{n}{\left\|e(k)-\omega (k)\right\|}^{2}\leq M^{\prime }<\infty$.*(3)* 
*The control strategy satisfies the stability requirements for the known objects of the parameters and tracks the reference.*
*(4)* *The controller parameter is a continuous function of the parameter estimate; that is,*$\theta_{c}(k)$*is a continuous function of*$\hat{\theta }(k)$.
Then, the self-tuning control system is stable and convergent. □

## 4. Extended Results

The above results can be extended to the colored noise situation. The difficulty is that the noise must be estimated together with the parameter estimation. 

Considering a general multivariate stochastic system ΣP.
(25)$$A(q^{-1})y(k)=q^{-d}B(q^{-1})u(k)+C(q^{-1})\omega (k),$$
where y(k), u(k), ω(k) has the same meanings as in (1).
$$\begin{array}{l} A(q^{-1})=I+A_{1}q^{-1}+\ldots +A_{n}q^{-n},n\geq 1\\ B(q^{-1})=B_{0}+B_{1}q^{-1}+\ldots +B_{m}q^{-m},m\geq 1\\ C(q^{-1})=I+C_{1}q^{-1}+\ldots +C_{r}q^{-r},r\geq 1 \end{array}$$

Introducing symbols
$$\theta^{T}=[-A_{1},\ldots ,-A_{n},B_{0},\ldots ,B_{m},C_{1},\ldots ,C_{r}],\;\phi_{0}{}^{T}(k-d)=[y(k-1),\ldots y(k-n),\ldots ,u(k-d),..\;u(k-d-m),\omega (k-1),\ldots ,\omega (k-r)].$$

Then, we have
$$y(k)=\phi_{0}{}^{T}(k-d)\theta +\omega (k).$$

The elements of the parameter matrix have changed to.
$${\hat{\theta }}^{T}(k)=[-{\hat{A}}_{1},\ldots ,-{\hat{A}}_{n},{\hat{B}}_{0},..,{\hat{B}}_{m},{\hat{C}}_{1},\ldots ,{\hat{C}}_{r}].$$

Since $\phi_{0}{}^{T}(k-d)$ contains unknown noise terms, it is necessary to estimate the noise terms while estimating the parameters so that the regression matrix (vector) of the parameter estimation is as follows.
$$\phi^{T}(k-d)=[y(k-1),\ldots y(k-n),\ldots ,u(k-d),..u(k-d-m),\hat{\omega }(k-1),\ldots ,\hat{\omega }(k-r)],$$
where
$$\hat{\omega }(k)=y(k)-\phi^{T}(k-d)\hat{\theta }(k).$$

The other symbols are the same as before. The self-tuning controller is denoted as ΣC(k), and can also be regarded as a matrix $\theta_{c}(k)$, which can be obtained by various control design methods. The self-tuning control system is abbreviated as (ΣC(k),ΣP), and the corresponding non-adaptive control system is abbreviated as (ΣC,ΣP). The virtual equivalent system of the self-tuning control system is abbreviated as $\left(\Sigma C,\Sigma P,\Delta u(k)\right)$, which can still be illustrated by Figure 3.

If the calculation of the control law does not require the estimation of noise, the calculation of u(k), Δu(k), $\phi_{c}(k)$ will not cause noise estimation problems.

If the calculation of the control law requires the estimation of noise, there will be the following problem. The regression matrices (vector) used to calculate u(k) and calculate $u_{0}(k)$ are different; that is,
$$u(k)=\phi_{c}^{T}(k)\theta_{c}(k)\text{,}\;u_{0}(k)=\phi_{c0}^{T}(k)\theta_{c}.$$
where
$$\phi_{c}{}^{T}(k)=[y_{r}(k),y_{r}(k-1)\ldots y(k),y(k-1)\ldots ,u(k-1),\ldots ,\hat{\omega }(k-1),\ldots ],\;\phi_{c0}{}^{T}(k)=[y_{r}(k),y_{r}(k-1)\ldots y(k),y(k-1)\ldots ,u(k-1),\ldots ,\omega (k-1),\ldots ].$$

However, due to condition (2) in Theorem 3.
$$\sum_{k=1}^{n}{\left\|e(k)-\omega (k)\right\|}^{2}=o(\alpha +\sum_{k=1}^{n}{\left\|\phi (k-d)\right\|}^{2}).$$

That is equivalent to (by definition, $\hat{\omega }(k)$ is e(k))
$$\sum_{k=1}^{n}{\left\|\overset{\wedge }{\omega }(k)-\omega (k)\right\|}^{2}=o(\alpha +\sum_{k=1}^{n}{\left\|\varphi (k-d)\right\|}^{2}).$$

Thus, the difference between $\phi_{c}(k)$ and $\phi_{c0}(k)$ can be merged into Δu(k) without affecting the property of Δu(k). Therefore, the above results (with white noise) still hold true for the general stochastic system (25).

**Remark** **2.**
*For simulation verification, see reference [47].*


## 5. Conclusions

Based on the equivalent system concept, a unified analysis of multivariable stochastic self-tuning control (STC) systems is presented. In this paper, by the virtual equivalent system, the difficulty of analyzing the stability and convergence of the stochastic self-tuning control system is transferred from the system structure to the compensational signal, which reduces the difficulty of the original problem, making the stability and convergence analysis of the stochastic self-tuning control system more intuitive and easier to understand. We investigated three situations, i.e., parameter estimation converges to the true value, parameter estimation converges to a non-true value, and parameter estimation may not converge.

## Figures and Tables

**Figure 1 entropy-25-00173-f001:**
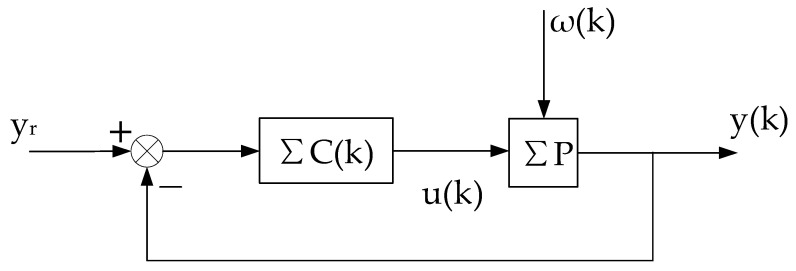
Stochastic self-tuning control system.

**Figure 2 entropy-25-00173-f002:**
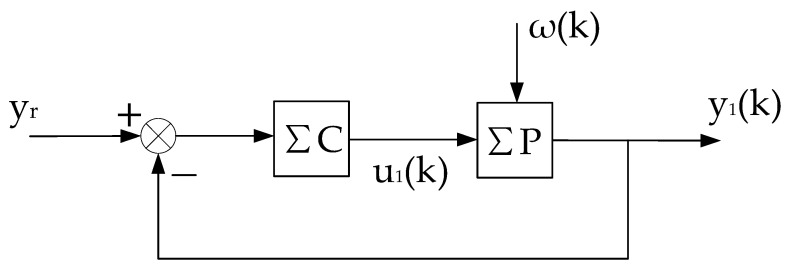
Constant control system corresponding to self-tuning control system.

**Figure 3 entropy-25-00173-f003:**
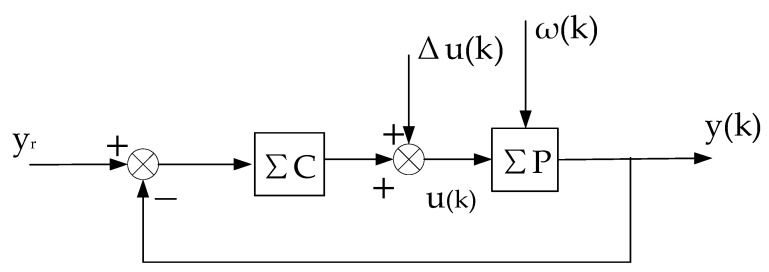
Virtual equivalent system of stochastic self-tuning control system.

**Figure 4 entropy-25-00173-f004:**
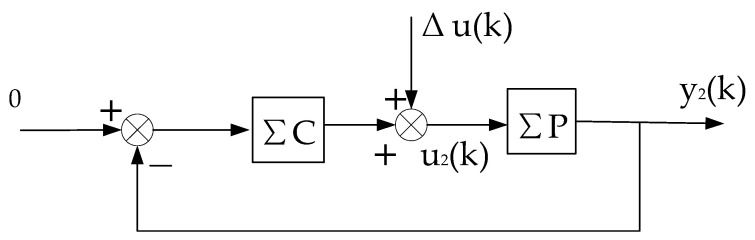
Decomposition subsystem of the virtual equivalent system 2.

**Figure 5 entropy-25-00173-f005:**
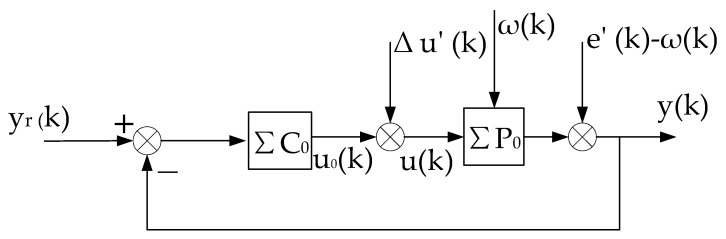
Virtual equivalent system when parameter estimates converge to non-true values.

**Figure 6 entropy-25-00173-f006:**
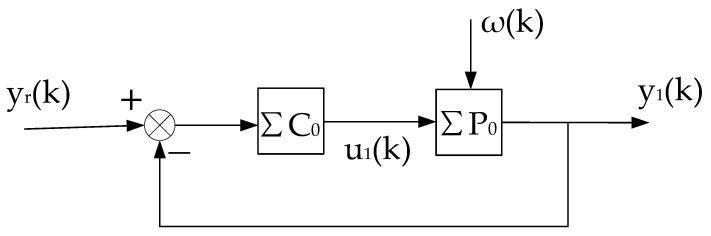
Decomposition system of virtual equivalent system 1.

**Figure 7 entropy-25-00173-f007:**
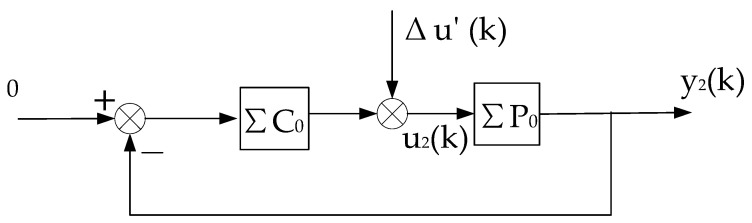
Decomposition system of virtual equivalent system 2.

**Figure 8 entropy-25-00173-f008:**
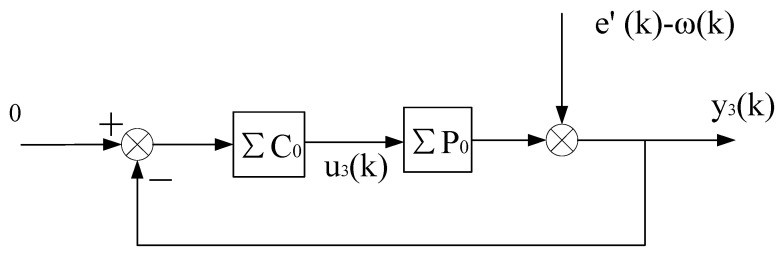
Decomposition system of virtual equivalent system 3.

**Figure 9 entropy-25-00173-f009:**
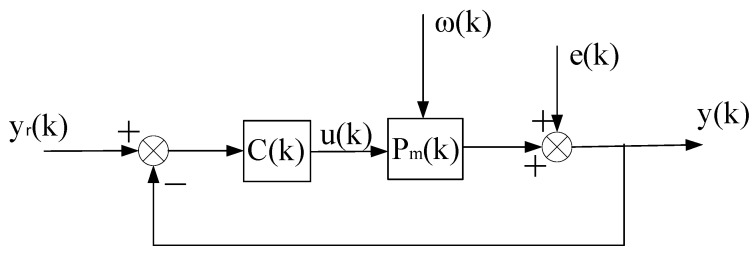
Virtual equivalent system I when parameter estimation may not converge.

**Figure 10 entropy-25-00173-f010:**
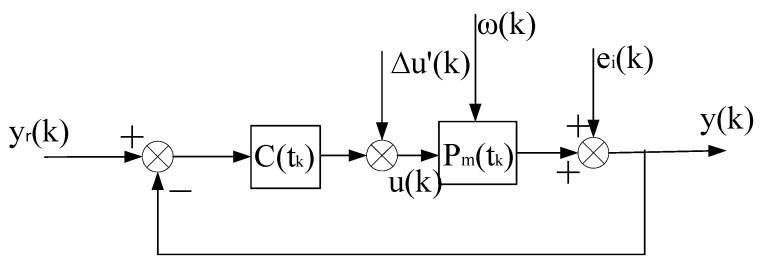
Virtual Equivalent System II when parameter Estimation may not convergence.

**Figure 11 entropy-25-00173-f011:**
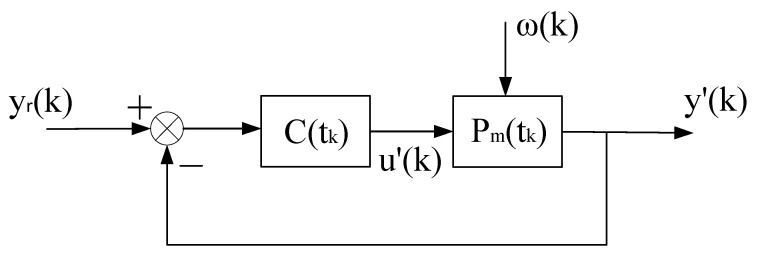
Decomposed subsystem 1.

**Figure 12 entropy-25-00173-f012:**
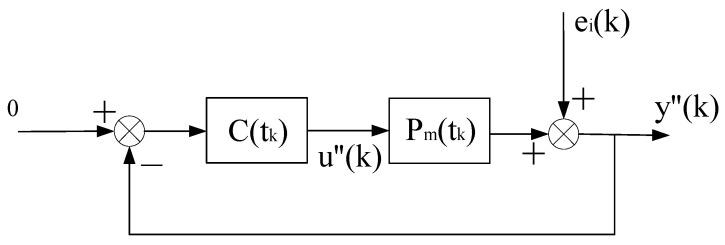
Decomposed subsystem 2.

**Figure 13 entropy-25-00173-f013:**
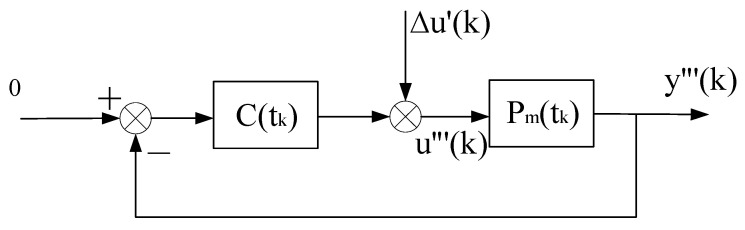
Decomposed subsystem 3.

## Data Availability

Data are contained within the article. The data presented in this study can be requested from the authors.

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
