# Peer review of "From Nonlinear Dominant System to Linear Dominant System: Virtual Equivalent System Approach for Multiple Variable Self-Tuning Control System Analysis"

_entropy, 2023, doi:10.3390/e25010173_

Round 1
Reviewer 1 Report
This paper describes the method to convert the multivariable stochastic self-tuning control(STC) system using virtual equivalent system approach. Parameter estimation was also covered for three cases. The idea introduced by the authors is quite interesting. However, there are still some issues that need to be improved.
- The Abstract needs to be organized again. Abstract does not represent or summarize concisely the problem to be covered in this study. Moreover, it is very hard to catch the contribution and novelty as explained in this manuscript.
- The stability and convergence are defined and given by two equations. (There were no equation numbers. So it is hard to indicate.) The two equations are written in front of Section 3. (The first 2 equations on top of page 4.) How can you guarantee that they are valid?
- Is a continuous-time system or discrete-time system adopted in this paper? y(k), u(k) was used instead of y(t), u(t), which makes readers confused if it is continuous-time system. If it is discrete-time, Schur stability should be used rather than Hurwitz stability.
- The proof of Theorem 1 & 2 are too long to make reader understand clearly. The procedures need to be written in systematic way.
- The background, motivation, and necessity of this study were not fully explained in the introduction section. With new reference papers, sufficient explanations should be added. In addition, a detailed discussion on the difference from other studies is required.
- A method to manifest the improvement of the proposed 4 theorems and 2 corollaries must be added. For example, a Section containing simulation results verifying the main results and extended results covered in Sections 3 and 4 should be added.
- The reference list should be updated with the latest references.
- In Figs. 6 to 8, all numbers should be treated as a subscript. For example, y_2(k) not y2(k).
- No period or comma mark after each equation.
- Most of all, there is no simulation results or experiments to prove the proposed method. This should be done for each theorem and the performance comparisons with other researcher’s previous result are required also.
Reviewer 2 Report
1. Will this transforming causes system modelling error unceretainty and how to quantitatively describe its impact on the stability or performance?
2. It is recommended to give essential simulation to evaluate the dynamics response of the system before and after transforming.
3. Compared to common methods, what is the superiority of the proposed method? Give essential analysis or simulation?
4. The investigation is not sufficient such as the classical nonlinear controller of sliding mode control 10.1109/TII.2021.3057832
5. The Introduction is not enclosed. The method, contribution and conclusion should be emphasized in more details.
Round 2
Reviewer 1 Report
The overall quality and completeness of the manuscript has been greatly improved compared to the previous submission.
Reviewer 2 Report
Thanks for the authors' response. However, some replies are not convincing and should be carefully considered.